# Optical Coherence Tomography Biomarkers in Predicting Treatment Outcomes of Diabetic Macular Edema after Ranibizumab Injections

**DOI:** 10.3390/medicina59030629

**Published:** 2023-03-21

**Authors:** Yen-Chieh Chang, Yu-Te Huang, Alan Y. Hsu, Ping-Ping Meng, Chun-Ju Lin, Chun-Ting Lai, Ning-Yi Hsia, Huan-Sheng Chen, Peng-Tai Tien, Jane-Ming Lin, Wen-Lu Chen, Yi-Yu Tsai

**Affiliations:** 1Department of Ophthalmology, China Medical University Hospital, China Medical University, Taichung 404333, Taiwan; 2Department of General Medicine, College of Medicine, China Medical University Hospital, Taichung 404332, Taiwan; 3School of Medicine, College of Medicine, China Medical University, Taichung 404333, Taiwan; 4Department of Optometry, Asia University, Taichung 40447, Taiwan; 5An-Shin Dialysis Center, NephroCare Ltd., Fresenius Medical Care, Taichung 43655, Taiwan; 6Graduate Institute of Clinical Medical Science, College of Medicine, China Medical University, Taichung 404333, Taiwan

**Keywords:** diabetic macular edema, disorganization of retinal inner layers, hyperreflective foci, intravitreal antivascular endothelial growth factor, ranibizumab, optical coherence tomography biomarkers, subretinal fluid, large outer nuclear layer cyst

## Abstract

*Background and Objectives*: The identification of possible biomarkers that can predict treatment response among DME eyes is important for the individualization of treatment plans. We investigated optical coherence tomography (OCT)-based biomarkers that may predict the one-year real-life outcomes among diabetic macular edema (DME) eyes following treatment by intravitreal ranibizumab (IVR) injections. *Materials and Methods:* A total of 65 eyes from 35 treatment-naïve patients with DME treated with ranibizumab injection were recruited. Best-corrected visual acuity (BCVA), central retinal thickness (CRT), intraocular pressure (IOP), and OCT scans were retrospectively recorded at baseline before treatment and at 3 months, 6 months, and 12 months after treatment. The OCT scans were evaluated for biomarkers of interest, which included central retinal thickness (CRT), amount and locations of hyperreflective foci (HRF), subretinal fluid (SRF), intraretinal cysts (IRC), large outer nuclear layer cyst (LONLC), ellipsoid zone disruption (EZD), disorganization of retinal inner layers (DRIL), hard exudates (HE), epiretinal membrane (ERM), and vitreomacular interface (VMI). Correlations between these OCT biomarkers and outcome measures (visual and structural) were statistically analyzed. *Results:* A total of 65 eyes from 35 patients with DME were enrolled. The mean age was 64.2 ± 10.9 years old. Significant improvement in terms of mean BCVA (*p* < 0.005) and mean CRT was seen at final follow-up compared to baseline. The biomarkers of DRIL, LONLC, and SRF were found to be predictive for at least 50 μm CRT reduction after treatment (with odds ratio of 8.69, 8.5, and 17.58, respectively). The biomarkers of IRC, LONLC, and SRF were predictive for significant improvement in terms of BCVA and CRT after treatment. Finally, the number of HRF was predictive for both BCVA improvement and a CRT reduction of less than 100 μm after treatment. No serious complications were reported during the study. *Conclusion:* Our study demonstrated the utility of OCT biomarkers as therapeutic predictors of ranibizumab treatment among DME eyes.

## 1. Introduction

Diabetic retinopathy (DR) is one of the complications of diabetic mellitus and is a leading cause of vision loss worldwide [1]. Its pathophysiology involves retinal microvascular leakage, which can result in diabetic macular edema (DME) and retinal hemorrhage. The severity of DR can be subdivided into either nonproliferative diabetic retinopathy (NPDR) or proliferative diabetic retinopathy (PDR) [2]. The management of DME involves intravitreal antivascular endothelial growth factor (anti-VEGF) injections, along with retinal photocoagulation and steroid treatments [3,4,5]. These treatments have been shown to improve visual acuity as well as prevent complications such as neovascular glaucoma, tractional maculopathy, or tractional retinal detachment [5]. 

Imaging tools such as optical coherence tomography (OCT) have become helpful adjuncts in everyday ophthalmology practice. OCT is a form of noninvasive imaging modality that provides cross-sectional images of the internal ophthalmologic structures. With tools such as this, it can be seen that the internal structures can contribute valuable information on various ophthalmologic conditions, including DME [6]. Anatomical measures from OCT such as central retinal thickness (CRT), disorganization of the retinal inner layers (DRIL), ellipsoid zone disruption (EZD), hard exudate (HE), hyperreflective foci (HRF), subretinal fluid (SRF), the presence of epiretinal membrane (ERM), and disruption of the vitreomacular interface (VMI) have been explored in the past for their predictive value among patients with DME about to undergo various therapies, but results have proven inconclusive so far [7,8,9,10,11,12,13,14,15,16,17,18,19,20,21,22]. Knowing which of these OCT parameters that can predict therapeutic response after intravitreal anti-VEGF injections would be helpful for clinicians in planning and individualizing treatment plans for patients [10,11,12,13,17,19,20,21,22]. This is especially important considering the fact that even though anti-VEGF therapies such as ranibizumab (Lucentis, Genentech Inc., South San Francisco, CA) are used to treat DME, the treatment response is not uniform. Studies have reported that 14.4% and 15.2% of patients exhibit nonresponse to treatment and worsening visual outcomes after IVR treatment [4,23].

Therefore, the aim of our study was to investigate whether certain characteristics found on OCT may serve as prognostic indicators for functional and structural outcomes after IVR treatment among DME patients using real-world data. 

## 2. Materials and Methods

This retrospective interventional case series was conducted at a single tertiary medical center of the China Medical University Hospital (CMUH) in Taiwan from January 2018 to January 2021. The study adhered to the tenets of the Declaration of Helsinki and was approved by the Institutional Review Board of CMUH (IRB number: CMUH109-REC3-158). Due to the retrospective study design, the requirement of informed consent was waived.

### 2.1. Inclusion and Exclusion Criteria

The inclusion criteria were as follows: (1) patients older than 18 years of age; (2) previous history of type 1 or 2 diabetes mellitus; (3) exhibiting of NPDR and PDR with leakage in the macular area documented by fluorescein angiography; (4) eyes possessing Snellen best-corrected visual acuity (BCVA) between 20/400 and 20/40 at initial presentation; (5) CRT on OCT greater than 300 μm at the initial presentation; (6) having received treatment with only intravitreal ranibizumab injections throughout the study duration; (7) having documented BCVA and OCT that were obtained at baseline before IVR treatment and at 3 months, 6 months, and 12 months after IVR injections. For patients with DME in two eyes and both having received injection, two eyes were recruited. 

The exclusion criteria were as follows: (1) the presence of retinal diseases other than DR that can cause macula edema; (2) previous history of any retinal surgery; (3) uncontrolled glaucoma; (4) any other ocular or systemic condition that can influence visual acuity; (5) PDR patients who received any additional laser photocoagulation during the study period. OCT images deemed poor-quality by the authors of this manuscript were also excluded.

Patients’ charts were reviewed for baseline demographics (e.g., gender, age etc.), type of retinopathy, and any previous treatments for DME. OCT, CRT and BCVA were recorded at baseline before IVR injections and at 3 months, 6 months, and 12 months after IVR injections. 

All patients received an initial three-month loading dose (0.5 mg) of IVR injections. When stable BCVA and CRT less than 300 µm were achieved, a treat-and-extend (T&E) regimen was then adopted, where the treatment intervals were increased by 4 weeks. However, if the patient had experienced vision loss or any other documented evidence of DME reoccurrence (CRT greater than 300 μm), then the IVR injection interval was reduced by 4 weeks at a time. If there was no recurrence, patients were allowed to extend their clinical visit and injection by one more month.

### 2.2. Imaging and Data Acquisition

The OCT data were acquired by Spectral domain OCT (SD-OCT) (Heidelberg Spectralis, Heidelberg, Germany). All SD-OCT images were macular scans centered on the fovea that consisted of 25 scans over a field of 6 × 6 mm area (Figure 1). Qualitative and quantitative evaluations of SD-OCT images were performed at baseline before IVR injections as well as 3 months, 6 months and 12 months after IVR. We assessed for the presence of OCT morphologic features of interest, which included (1) hyperreflective foci (HRF); (2) subretinal fluid (SRF); (3) intraretinal cyst (IRC); (4) large outer nuclear layer cyst (LONLC); (5) ellipsoid zone disruption (EZD); (6) disorganization of retinal inner layers (DRIL); (7) hard exudate (HE); (8) epiretinal membrane (ERM); and (9) vitreomacular interface (VMI). 

Morphological parameters of each OCT biomarker of interest are described below. Hyperreflective foci (HRF) are defined as the presence of well-circumscribed dots within retina of similar reflectivity as the retinal pigmental epithelial (RPE) band on OCT. Subretinal fluid (SRF) is the presence of hyporeflective areas between the sensory retina and RPE. Intraretinal cyst (IRC) was defined as hyporeflective spaces located within the sensory retina. Large outer nuclear cyst (LONC) was defined as hyporeflective spaces with size over 100 μm located within the outer nuclear layer. Ellipsoid zone disruption (EZD) was defined as any disruption in integrity noted within the ellipsoid zone. Disorganization of the retinal inner layers (DRIL) was defined as the presence of any disruption in the integrity of the retinal inner layers. Hard exudate (HE) was defined as the presence hyperreflective foci with prominent shadowing effect on OCT within the retina. Epiretinal membrane (ERM) was defined as a hyperreflective layer on the inner surface of the retina. Finally, disruption in the vitreomacular interface (VMI) was defined as the presence of any disruptions in the vitreomacular interface, including any posterior vitreous detachment (PVD), vitreomacular adhesion, and vitreomacular traction.

Manual evaluation of the obtained OCT images was performed by two evaluators (Y.-C.C. and Y.-T.H.). Details of the patient’s clinical findings and systemic parameters were masked throughout the OCT evaluation process. If any disagreement on the grading was encountered, then a third senior retina specialist (C.-J.L.) would make the final decision. Computer-aided analysis of OCT images by OCT software was also performed to evaluate for other anatomical measures such as CRT.

### 2.3. Statistical Analysis

All analyses were computed by using PASW Statistics 18 software (Version 18.0. SPSS Inc.: Chicago, IL, USA). The quantitative data were reported as arithmetic mean and standard deviation for continuous variables in the text and Tables. Categorical variables were reported as absolute frequency and calculated percentages. The baseline clinical status of patients and changes in CRT and BCVA were analyzed using Chi-square and one-way ANOVA. Multivariate logistic regression and general linear model analyses were utilized to evaluate the presence of several OCT biomarkers (HRF, SRF, IRC, LONLC, EZD, DRIL, HE, ERM, VMI) as predictive factors for final visual acuity improvement at the end of treatment. McNemar’s test was used to compare change in positive biomarker cases between baseline and study end for each outcome subgroup. A *p*-value less than 0.05 was considered to be statistically significant.

## 3. Results

### 3.1. Study Population

A total of 65 eyes from 35 patients were included in this analysis. Demographics and baseline characteristics are shown in Table 1. The mean age was 61.7 ± 10.6 years old. A total of 13 (37%) of 35 patients were female and 22 (63%) of the patients were male. The mean value of HbA1c levels was 8.9 ± 2.5%. The average numbers of IVR injection were 7.6 ± 2.4. There was 1 eye (1.5%) with severe NPDR, while 42 (64.6%) had PDR. A total of 22 eyes (33.9%) had PDR that had previously received panretinal photocoagulation (PRP) treatment. The mean BCVA at baseline was 0.5 ± 0.4 logMAR, the mean IOP at the baseline was 16.0 ± 3.5 mmHg, and the mean CRT at the baseline was 361.4 ± 99.7 µm.

### 3.2. Anatomical and Functional Outcome

There was a significant reduction (*p* < 0.0001) in CRT for all 65 eyes after IVR treatments at final follow-up at the 12th month compared to baseline. The baseline mean initial CRT was 361.42 ± 99.72 µm and mean CRT at the 12-month follow-up after IVR treatments was 300.02 ± 47.03 µm (Figure 2A). During the follow-up period, CRT showed a significant reduction in the first 3 months, and this trend was sustained until the end of the study period (Figure 2B). The mean BCVA change in LogMAR for all 65 eyes after IVR treatment also showed a significant improvement (*p* < 0.005) at final follow-up compared to baseline before treatment. The baseline BCVA was 0.51 ± 0.41, compared to the 12-month BCVA of 0.38 ± 0.48 (Figure 3A). A gradual trend towards BCVA improvements was noted initially and only reached statistical significance at the 12th month of follow-up after IVR (Figure 3B). BCVA was noted to take longer in achieving statistical significance after follow-up compared to CRT (statistical significance was achieved at the 12th month for BCVA versus the 3rd month for CRT, respectively). The overall changes in mean IOP also achieved statistical significance at the final-follow up compared to baseline (baseline IOP 16.02 ± 3.52 compared to 12-month IOP of 15.25 ± 2.87, *p* < 0.05) (Figure 4A). No significant elevation in IOP was measured throughout the study period among all our study participants. No serious complications were reported during the study.

### 3.3. Optical Coherence Tomography Biomarker Analysis

In terms of baseline OCT biomarkers, the age group of patients younger than 65 years old was associated with a higher number of HRF (96.9%) and VMI (87.8%) at baseline, with a majority of HRF involving all layers (65.6%). A Chi-square test was conducted to determine if any association exists between initial BCVA and CRT with the presence of OCT biomarkers at baseline. Our results showed that those with poor initial BCVA (defined as LogMAR greater than 0.4) were more likely to be associated with baseline OCT markers of DRIL, EZD, HE, and SRF, as well as HRF numbering over 20 and HRF involving all layers. We also showed that those with worse CRT (>350 μm) at baseline were more likely to be associated with baseline OCT biomarkers of DRIL, ERM, EZD, IRC, and LONLC (Table 2).

Multiple regression analysis was also conducted and showed that eyes presenting with DRIL, LONLC, and SRF at baseline OCT were associated with a CRT improvement of more than 50 μm at final follow-up (with vs. without DRIL, LONLC, and SRF: odds ratio 8.69, 8.50, and 17.58, respectively) (Table 3). General linear modeling also confirmed association between these OCT biomarkers (DRIL, LONLC, SRF) and CRT improvement after IVR treatment (model adjusted R-square 0.49, *p* = 0.0016) (Table 4). However, no OCT biomarkers at baseline were found to be predictive for BCVA improvement after IVR treatment.

We noted a significant reduction in IRC (64.6% to 49.2%), LONLC (41.5% to 21.5%), SRF (13.8% to 1.5%), and number of HRF after IVR treatment (*p* < 0.05) compared to baseline (Table 5). Separate subgroup analysis was also performed to identify any associations between presence of OCT biomarkers and changes in CRT and BCVA at different time points within the study period. In one subgroup analysis, DRIL (100% to 64.2%), LONLC (100% to 42.8%), and SRF (42.8% to 7.1%) were found to be significantly associated with final CRT improvements greater than 100 μm. Interestingly, none of these OCT biomarkers (DRIL, LONLC, SRF) were significantly correlated with CRT improvements less than 100 μm. In the subgroup analysis classified by BCVA response, reductions in HRF number, LONLC (34.6% to 16.3%), and SRF (14.2% to 2.0%) OCT biomarkers were found to be significantly associated with BCVA improvement (ΔLogMAR ≤ 0) after IVR treatment. 

## 4. Discussion

Our study sought to comprehensively evaluate several OCT biomarkers of interest in a single study and its association with visual and structural outcomes after ranibizumab treatment in eyes with DME.

### 4.1. Specific Findings 

Our findings showed that the presence of the OCT biomarkers of DRIL, LONLC, and SRF at baseline were correlated with a CRT improvement of more than 50 μm after ranibizumab treatment among DME eyes. Within the subgroup analysis based on CRT improvements, OCT biomarkers of DRIL, LONLC, and SRF were all found to be significantly associated with a greater than 100 μm reduction in final CRT compared to baseline. Within the other subgroup analysis classified by BCVA improvements, a decrease in the number of HRF was correlated with BCVA improvements after IVR treatment. 

### 4.2. Clinical Implications

There are few studies to date that sought to investigate the predictive value of several SD-OCT biomarkers in terms of treatment response after ranibizumab injections under real-world conditions. Our study demonstrated certain OCT features to be significantly associated with early functional and structural response in DME eyes after IVR. Our results are important as they contribute towards the literature about the potential usage of such parameters. Once validated in future studies, these parameters have the potential to transform clinical care for DME patients through its theoretical applications towards active monitoring as well as influencing the treatment plans for such clinical cohort. 

### 4.3. Comparisons to Other Studies 

Certain OCT features, including HE, HRF, IRC, LONLC, SRF, and DRIL, which are commonly seen among DME eyes [7,8,9,10,11,12,13,14,15,16,17,18,19,20,21,22] have been reported to be potentially predictive towards response to certain intraocular treatment [10,12,13,17,19,20,21,22]. We seek to explore these associations within our study and prove their validity. 

To begin with, our results showed that the baseline presence of DRIL, LONLC, and SRF on OCT among DME eyes was significantly associated with CRT improvement by more than 50 µm after ranibizumab treatment (Table 3). We chose to focus on these biomarkers as studies such as Sophie et al. have qualified these markers to be predictive in terms of both a reduction in the final CRT as well as the BCVA obtained at final follow-up (OR, 2.88, *p* = 0.0001) [8]. 

Intraretinal cysts (IRC) were also another OCT parameter that was explored in our study. This biomarker was chosen to be studied as previous randomized control trials such as Szeto et al. have shown IRCs to be significantly associated with a change in CST (*p* < 0.001) [24]. From our study, we further corroborate this by finding IRCs to be significantly associated with a reduction in CRT at final follow-up compared to baseline after treatment with IVR. Again, this is another important finding, as real-world data were provided from our study that hint towards the utility of IRC in predicting CRT reductions post-IVR treatment.

Subretinal fluid (SRF) was the next biomarker of interest. SRF is where there is fluid accumulation in subretinal space and is thought to occur secondary to the disrupted blood–retinal layer of the outer retina as well as RPE dysfunction, leading to excess extracellular fluid accumulation [18,25]. This OCT biomarker, chosen in a previous report, has shown to be predictive towards functional and anatomic outcomes at final follow-up after ranibizumab treatment (OR, 2.43; *p* = 0.004) [8]. These were again consistent with our results, as we have shown SRF to be predictive for CRT improvement. The reasons for the association between SRF and improved structural and functional outcomes are still unclear. One possible explanation for this is that the presence of fluid between the retina and the choriocapillaris (i.e., subretinal fluid) may serve as a physical barrier that prevents any further deterioration of the retinal anatomical structures. The subretinal fluid itself may also contain protective substances and this protective function may be further enhanced by the addition of anti-VEGF injections. Therefore, it is possible that based on these reasons, eyes containing subretinal fluid may have better outcomes when treated with anti-VEGF injections. This was hinted in studies where eyes with subretinal fluid were at lower risk of developing geographic atrophy than eyes without subretinal fluid (adjusted hazard ratio 0.52) [25]. Despite unanswered questions on the exact pathophysiology, our data on SRF are again another important finding with real-world implications as they highlight its potential as an OCT biomarker in predicting post-IVR treatment outcomes. 

Hyperreflective foci (HRF) have also been suggested in previous literature to reflect the inflammatory pathophysiology that underlines DME as well as correlating with disease severity among DME eyes [7,12,19]. This biomarker of interest was looked into by our study, as a previous report based on 54 eyes showed a significant reduction in CRT to be associated with a higher number of HRF at baseline (estimated effect −2.61, *p* = 0.006) [21]. Schreur et al. also showed that eyes with good treatment response to anti-VEGF injections had more HRF at baseline (OR 1.106, *p* = 0.03) compared to eyes with poor response. However, HRF was also shown to not be associated with changes in BCVA at the 3rd month of follow-up after anti-VEGF treatment by Schreur et al. These results partially complement our own. One major difference was that from our subgroup analysis, we showed that the number of HRF seen at baseline is significantly associated with better visual outcome after treatment. However, it should be noted that this relationship between HRF and improved visual outcome was only demonstrated within our subgroup analysis. The other statistical analysis employed by our study failed to replicate this correlation between HRF and final VA outcome. This possibly implies that the statistical strength was not strong and that further studies are still needed to validate our findings. Another possible explanation for our contradictory results was that in actual fact, it is not the HRF itself that are causing the visual changes, but rather, a nonspecific and ongoing inflammatory state that is affecting the retinal microstructure. Thus, when DME eyes receive anti-VEGF treatment, this inflammatory process would then be suppressed. When this happens, it would then result in an improvement in visual acuity [26].

Another OCT biomarker of interest was CRT. This biomarker was explored as studies such as Santos et al. [14] have demonstrated that a reduction in CRT is associated with improvement in BCVA (odds ratio 3.31). In contrast, no correlation between CRT improvement and BCVA recovery was found in our study. We hypothesize the following explanations for this. Firstly, it is possible that the tertiary nature of our hospital could have introduced a form of selection bias into our results. This means that there would be a tendency to recruit patients who are at a more severe stages of PDR. This was hinted in our baseline characteristics, where our mean HbA1c level was 8.9%, and 98.5% of our patients possessed characteristic neovascularization changes of PDR at baseline. Despite our lack of findings, our results are still important as they provide valuable real-world data from which methodologies of future studies can build upon. It is possible that future studies of better design may produce a different result than the one we obtained for CRT and its predictive value. 

No relation towards functional and anatomical prognosis was found for the other OCT parameters such as ERM, EZD, HE, and VMI. We chose these parameters to study as some reports have found some correlations. Gao et al. reported that patients with VMI disruption at baseline tend to have decreased response to anti-VEGF agents [27]. EZD and ERM integrity were also reported to be predictive towards final BCVA outcomes among eyes with macular edema in one study [28]. In terms of HE, this OCT parameter was chosen as it is a frequently observed phenomenon among DME eyes and was reported to reduce after anti-VEGF treatment in one study [29]. Our lack of findings in terms of any associations between these OCT parameters and treatment response to IVR is important, as it would not only inform future studies on these parameters but also inform clinicians to possibly focus their attention on other OCT parameters with more potential. It should be also highlighted that due to the overall heterogeneity of available studies on these different OCT parameters, we are unable to make direct comparisons between our studies and others. Furthermore, most studies have been retrospective in nature, and therefore, definitive causality cannot be made either. Future studies with prospective design and larger cohorts are still needed before any definitive conclusions can be made.

It should be also noted that systemic factors and medication history could also have an impact on our results. As numerous studies will show, DME, along with lipid metabolism disorders and renal dysfunction, all share similarities in terms of etiopathogenesis that revolves around hypoxia and inflammation. Hypoxia and inflammation are also central to the development of some OCT parameters such as HRF and HE. Researchers have hypothesized that HRF and HE are actually aggregates of microglial cells within the retinal layers that would manifest secondary to hypoxic conditions produced by either the dysfunctional retina or concomitant systemic disorders [30]. Therefore, OCT biomarkers such as HRF and HE could be a reflection of ongoing systemic disorders (such as chronic kidney disease or lipid metabolism disorders) rather than just a simple marker of DME activity. This has implications for our study, as we did not account for other underlining systemic disorders and how it influenced the manifestations of OCT parameters at baseline and at subsequent follow-ups. Furthermore, systemic disorders can also influence the responsiveness of retinal structures to anti-VEGF therapies. In one study, lower levels of certain lipid markers were associated with larger CRT reduction after anti-VEGF treatment [31]. Furthermore, the same study also reported that those with renal dysfunction tend to have poorer visual gains after anti-VEGF treatment. Medication use by patients related to the treatment of systemic disorders deserves some discussion as well. Dysfunctional serum lipids play a role in the pathophysiology of DME, as mentioned earlier. Statins, a drug commonly used to treat hyperlipidemia, have antioxidant properties that can reduce reactive oxygen species and prevent retinal neovascularization. Therefore, it is reasonable to theorize that the use of medication such as statins can have an effect on IVR treatment outcomes among our individual patients [32]. In summary, we highlight the possible effects that systemic disorders and the related medication history would have for our study. Unfortunately, we did not account for these factors within our baseline characteristics obtained. This is something that future studies can further explore. 

Another interesting finding from our study was with regard to IOP levels. Our results showed a statistical reduction in IOP after treatment with IVR among DME eyes. Literature on IOP changes after anti-VEGF injections have been mixed so far. Several studies have reported sustained IOP elevations after intravitreal injection, while others reported no long-term changes in IOP [33]. As previous studies mostly focus on IOP elevation after intravitreal steroid injection, we suggest that our results might be an incidental finding. 

### 4.4. Strengths and Limitations

One of the primary strengths of our study was that our data were based on real-world settings seen among ophthalmologic hospitals in Taiwan. Furthermore, our study had comprehensively investigated the predictive value of multiple SD-OCT biomarkers in terms of visual and anatomical outcomes after ranibizumab treatment among DME eyes. 

The main limitations of our study, however, include its retrospective design, a relatively small sample size, and short follow-up period. Due to our study’s retrospective characteristic, we can only infer association between baseline characteristics (OCT parameters) and treatment outcomes with no definitive conclusions on causality that can be made. Furthermore, due to our hospital being a tertiary medical center for ophthalmology, this introduces potential selection bias into our results.

### 4.5. Conclusions

In conclusion, we provided real-world data on OCT parameters that can potentially predict the therapeutic response after IVR injections among DME eyes. Specifically, we have found that the presences of DRIL, LONLC, IRC, and SRF were significantly associated with CRT improvement after IVR treatment. Additionally, a reduction in HRF was also found to predict BCVA improvement after treatment. The OCT biomarkers of DRIL, SRF, LONLC, IRC, and HRF therefore have potential in predicting the treatment response after IVR treatment. Further large-scale studies are still needed to validate our findings.

## Figures and Tables

**Figure 1 medicina-59-00629-f001:**
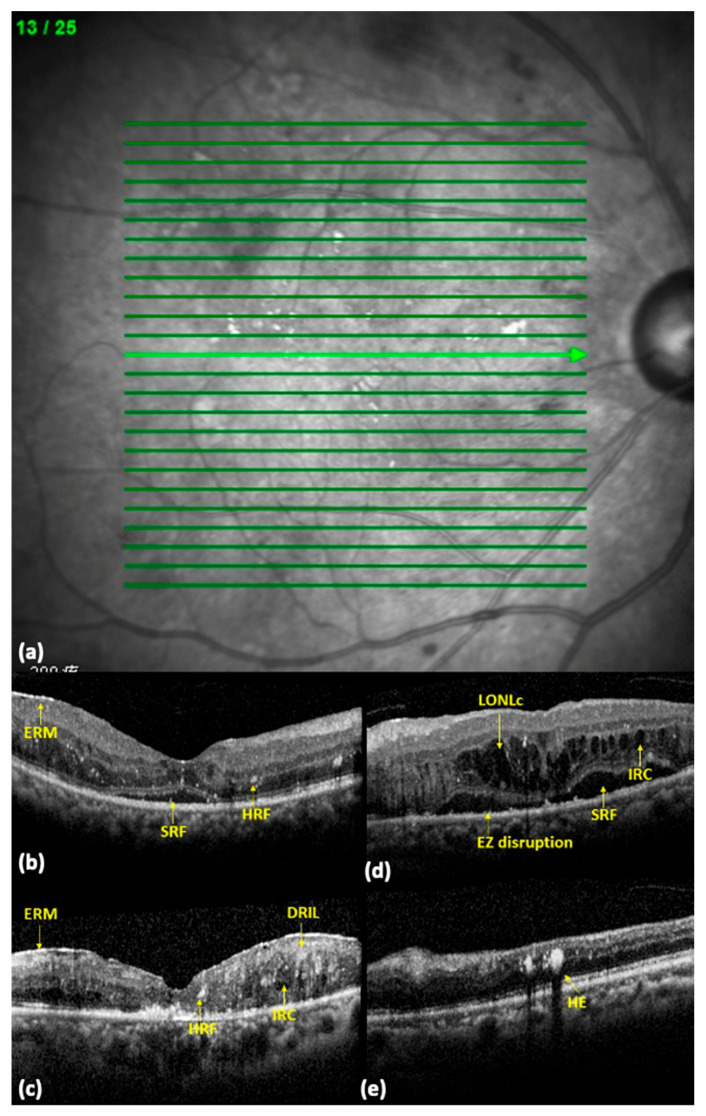
OCT biomarkers in diabetic macular edema (DME) eyes. (**a**) A representative image of a 25-6 mm macular scans centered on the fovea by spectral domain OCT (SD-OCT). (**b**) An OCT of a patient with DME showing the presence of epiretinal membrane (ERM), subretinal fluid (SRF), and hyperreflective foci (HRF) marked with yellow arrows. (**c**) An OCT of a patient with DME showing the presence of ERM, HRF, intraretinal cyst (IRC), and disorganization of the retinal inner layers (DRIL). (**d**) An OCT of a patient with DME showing the presence of large outer nuclear layer cyst (LONLC), ellipsoid zone disruptions (EZD), SRF, and IRC. (**e**) An OCT of a patient with DME showing the presence of hard exudate (HE).

**Figure 2 medicina-59-00629-f002:**
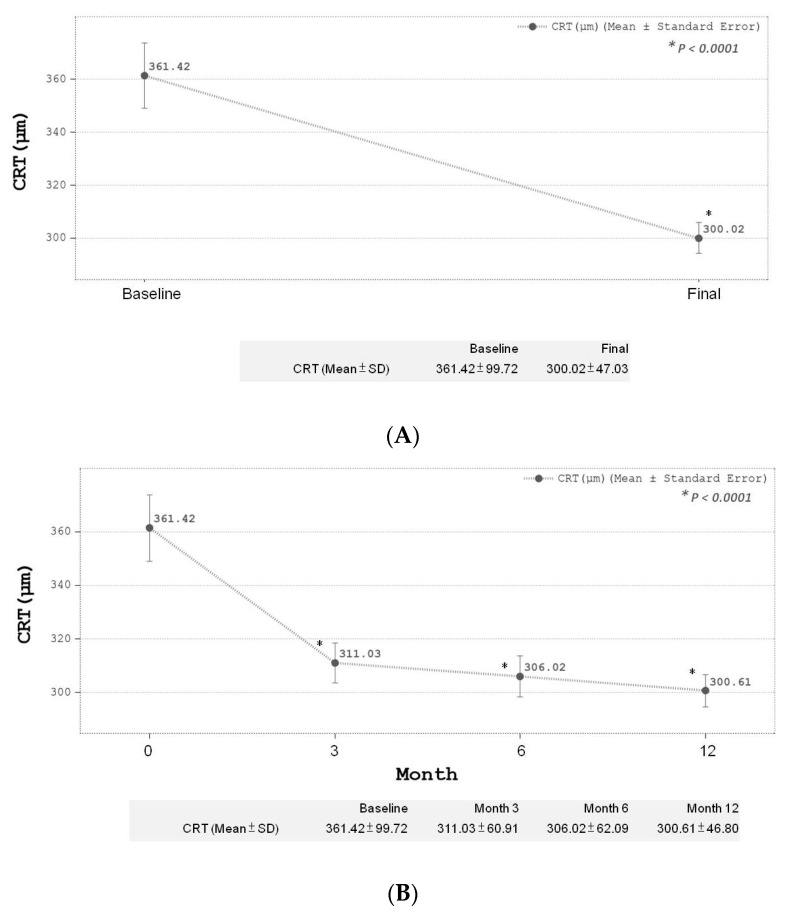
(**A**) Final CRT change after treatment. (**B**) Serial CRT change during treatment.

**Figure 3 medicina-59-00629-f003:**
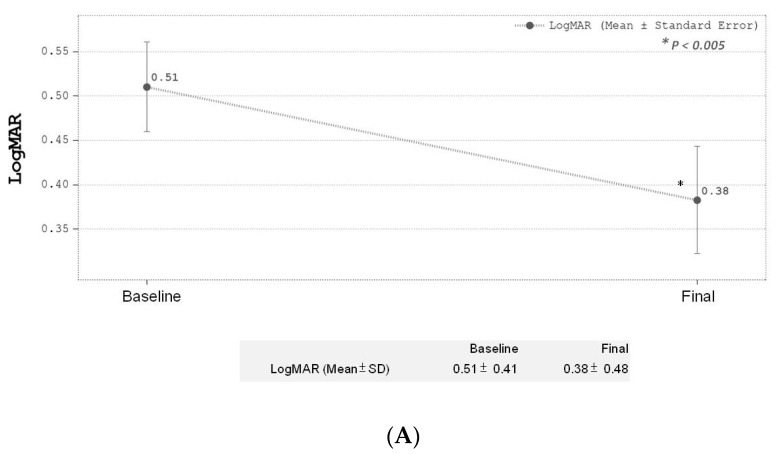
(**A**) Final BCVA (LogMAR) change after treatment. (**B**) Serial BCVA change during treatment.

**Figure 4 medicina-59-00629-f004:**
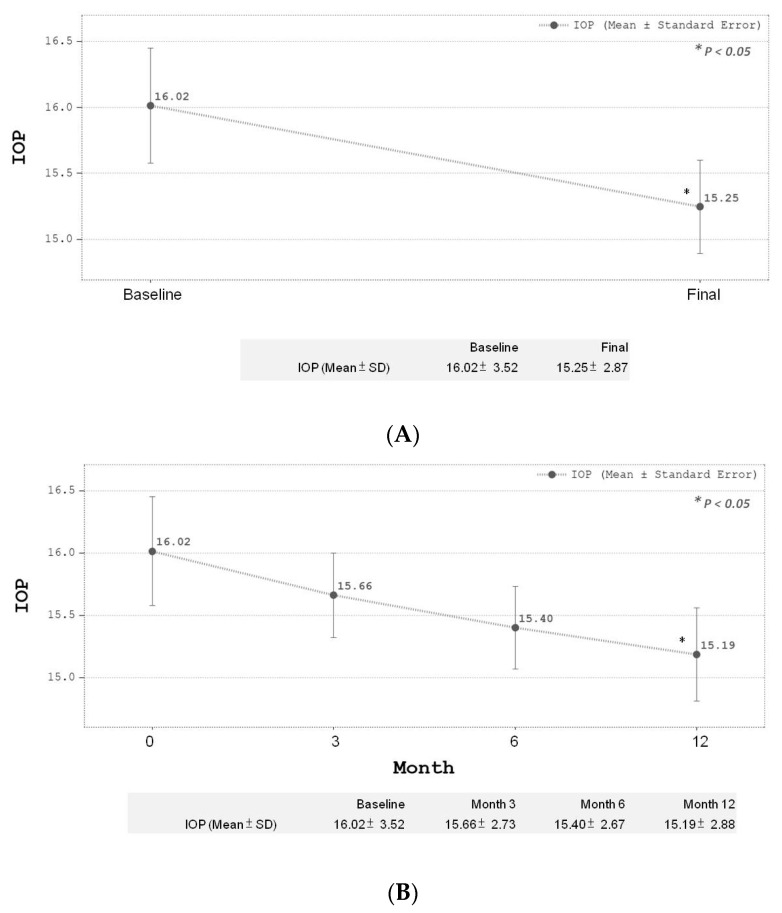
(**A**) Final IOP change after treatment. (**B**) Serial IOP change during treatment.

**Table 1 medicina-59-00629-t001:** Baseline demographics.

**Patients (No)**	35
Age, yrs (mean ± sd)	64.2 ± 10.9
Gender, female/total (%)	13/35 (37.1%)
HbA1c, % (mean ± sd)	8.9 ± 2.5
**Eyes (No)**	65
Sides, OD/total, (%) Numbers of IVR injection (mean ± sd)	33/65 (50.8%) 7.6 ± 2.4
Lens, pseudophakic/total, (%)	35/65 (53.9%)
Initial CRT	361.4 ± 99.7
Initial LogMAR	0.5 ± 0.4
Initial IOP	16.0 ± 3.5
Diabetic retinopathy	
Severe NPDR	1/65 (1.5%)
PDR	42/65 (64.6%)
PDR s/p PRP	22/65 (33.9%)

**Table 2 medicina-59-00629-t002:** Baseline biomarker status, stratified by baseline clinical parameters (eyes with positive markers/total eyes, %).

Biomarkers (No)	Whole	Age			Initial BCVA (LogMAR)			Initial CRT (μm)		
		≤65	>65	*p ^2^*	≤0.4	>0.4	*p*	≤350	>350	*p*
DRIL (65)	34 (52.3%) ^1^	18 (54.5%)	16 (50.0%)	0.7138	12 (33.3%)	22 (75.8%)	* 0.0006	14 (35.8%)	20 (76.9%)	* 0.0012
ERM (65)	32 (49.2%)	17 (51.5%)	15 (46.8%)	0.7083	16 (44.4%)	16 (55.1%)	0.3898	15 (38.4%)	17 (65.3%)	* 0.0334
EZD (65)	50 (76.9%)	23 (69.6%)	27 (84.3%)	0.1603	22 (61.1%)	28 (96.5%)	* 0.0007	26 (66.6%)	24 (92.3%)	* 0.0162
HE (65)	29 (44.6%)	15 (45.4%)	14 (43.7%)	0.8901	10 (27.7%)	19 (65.5%)	* 0.0023	14 (35.8%)	15 (57.6%)	0.0833
HRF (65)	56 (86.2%)	32 (96.9%)	24 (75.0%)	* 0.0104	30 (83.3%)	26 (89.6%)	0.4632	33 (84.6%)	23 (88.4%)	0.6601
Number of HRF (56)				0.2764			* 0.0143			0.2095
2–10	17 (30.4%)	8 (25.0%)	9 (37.5%)		12 (40.0%)	5 (19.2%)		13 (39.3%)	4 (17.3%)	
10–20	18 (32.1%)	13 (40.6%)	5 (20.8%)		12 (40.0%)	6 (23.0%)		9 (27.2%)	9 (39.1%)	
>20	21 (37.5%)	11 (34.3%)	10 (41.6%)		6 (20.0%)	15 (57.6%)		11 (33.3%)	10 (43.4%)	
HRF location (56)				* 0.0071			* 0.0407			0.9981
All layers	29 (51.8%)	21 (65.6%)	8 (33.3%)		13 (43.3%)	16 (61.5%)		17 (51.5%)	12 (52.1%)	
OPL-ONL	22 (39.3%)	11 (34.3%)	11 (45.8%)		16 (53.3%)	6 (23.0%)		13 (39.3%)	9 (39.1%)	
ILM-INL	5 (8.9%)	0 (0.0%)	5 (20.8%)		1 (3.33%)	4 (15.3%)		3 (9.0%)	2 (8.6%)	
IRC (65)	42 (64.6%)	23 (69.6%)	19 (59.3%)	0.3843	20 (55.5%)	22 (75.8%)	0.0888	21 (53.8%)	21 (80.7%)	* 0.0262
LONLC (65)	27 (41.5%)	15 (45.4%)	12 (37.5%)	0.5153	12 (33.3%)	15 (51.7%)	0.1347	6 (15.3%)	21 (80.7%)	* <0.0001
SRF (65)	9 (13.9%)	4 (12.1%)	5 (15.6%)	0.6826	1 (2.7%)	8 (27.5%)	* 0.0040	3 (7.6%)	6 (23.0%)	0.0785
VMI (65)	41 (63.1%)	29 (87.8%)	12 (37.5%)	* <0.0001	24 (66.6%)	17 (58.6%)	0.5040	23 (58.9%)	18 (69.2%)	0.4012

^1^ Eyes with positive markers (%). ^2^ Chi-square test, comparing % of positive biomarker cases in two different conditions for each clinical parameter. Outer plexiform layer, OPL; outer nuclear layer, ONL; internal limiting membrane, ILM; internal nuclear layer, INL. * indicates that the p value is significant for each table (*p* < 0.05).

**Table 3 medicina-59-00629-t003:** OCT Biomarkers as outcome predictors by multivariate logistic regression with final CRT improved ≥ 50 μm or not as dependent variable *.

Logistic Regression, Dependent Variable: Final CRT Improved ≥ 50 μm							
Parameter	Estimate	Standard Error	Wald Chi-Square	Pr > ChiSq	Odds Ratio	95% Confidence Limits	
Intercept	0.3352	0.6867	0.2383	0.6254			
DRIL (+)	1.0813	0.3913	7.6356	0.0057	8.694	1.875	40.311
LONLC (+)	1.0705	0.3695	8.396	0.0038	8.509	1.999	36.211
SRF (+)	1.4335	0.6864	4.3613	0.0368	17.586	1.193	259.272

* Dependent variables included before model selection: gender, age, lens status, and all baseline OCT biomarkers.

**Table 4 medicina-59-00629-t004:** OCT biomarkers as outcome predictors by general linear model with extent of CRT improvement (ΔCRT) after treatment as dependent variable *.

		Elastic Net Selection Summary						
Effect	Step	Model R-Square	Adjusted R-Square	Estimate	AIC	BIC	F Value	Pr > F
Intercept		0	0	−10.88288	660.9025	594.4552	0	1
LONLC (+)	1	0.1235	0.1096	−13.1763	654.3362	587.3349	8.87	0.0041
SRF (+)	2	0.4322	0.4138	−77.73782	628.1173	562.5221	33.71	<0.0001
DRIL (+)	3	0.5179	** 0.4942	−81.85528	** 619.4722	** 554.9817	10.85	0.0016

* Dependent variables included before model selection: gender, age, lens status, and all baseline OCT biomarkers. ** Optimal value of criterion.

**Table 5 medicina-59-00629-t005:** Biomarker changes after study, stratified by different outcome groups.

Biomarkers (No)	Whole			Final CRT Response							Final BCVA Response				
				CRT Improved ≥ 100 μm (ΔCRT ≤ −100 μm)			CRT Improved ≥ 100 μm (ΔCRT ≤ −100 μm)			BCVA Improved (ΔLogMAR ≤ 0)			BCVA not Improved (ΔLogMAR > 0)		
	Baseline	End	*p* ^2^	Baseline	end	*p*	Baseline	end	*p*	Baseline	end	*p*	Baseline	end	*p*
DRIL (65)	34 (52.3%) ^1^	28 (43.0%)	0.1336	14 (100.0%)	9 (64.2%)	* 0.0136	20 (39.2%)	19 (37.2%)	0.7630	23 (46.9%)	19 (38.7%)	0.2482	10 (71.4%)	8 (57.1%)	0.3173
ERM (65)	32 (49.2%)	37 (56.9%)	0.2253	8 (57.1%)	7 (50.0%)	0.6547	24 (47.0%)	30 (58.8%)	0.0833	21 (42.8%)	31 (63.2%)	* 0.0039	9 (64.2%)	6 (42.8%)	0.0833
EZD (65)	50 (76.9%)	45 (69.2%)	0.1655	14 (100.0%)	11 (78.5%)	0.0668	36 (70.5%)	34 (66.6%)	0.5271	34 (69.3%)	32 (65.3%)	0.5271	14 (100.0%)	12 (85.7%)	0.1422
HE (65)	29 (44.6%)	25 (38.4%)	0.3458	8 (57.1%)	11 (78.5%)	0.1797	21 (41.1%)	14 (27.4%)	0.0522	22 (44.8%)	18 (36.7%)	0.2482	6 (42.8%)	7 (50.0%)	0.6547
HRF (65)	56 (86.1%)	59 (90.7%)	0.3173	13 (92.8%)	13 (92.8%)	1.0000	43 (84.3%)	46 (90.1%)	0.2568	43 (87.7%)	45 (91.8%)	0.4142	11 (78.5%)	13 (92.8%)	0.1573
Number of HRF (56)			* 0.0148			0.8089			* 0.0078			* 0.0107			0.7602
2–10	17 (30.3%)	30 (50.8%)		1 (7.6%)	2 (15.3%)		16 (37.2%)	28 (60.8%)		11 (25.8%)	23 (51.1%)		4 (36.3%)	6 (46.1%)	
10–20	18 (32.1%)	20 (33.8%)		6 (46.1%)	6 (46.1%)		12 (27.9%)	14 (30.4%)		14 (32.5%)	15 (33.3%)		4 (36.3%)	5 (38.4%)	
>20	21 (37.5%)	9 (15.2%)		6 (46.1%)	5 (38.4%)		15 (34.8%)	4 (8.6%)		18 (41.8%)	7 (15.5%)		3 (27.2%)	2 (15.3%)	
HRF location (56)			0.4437						0.1314			0.3846			0.9418
All layers	29 (51.7%)	24 (40.6%)		7 (53.8%)	9 (69.2%)	0.3928	22 (51.1%)	15 (32.6%)		25 (58.1%)	20 (44.4%)		4 (36.3%)	4 (30.7%)	
OPL-ONL	22 (39.2%)	30 (50.8%)		5 (38.4%)	2 (15.3%)	0.3928	17 (39.5%)	28 (60.8%)		14 (32.5%)	21 (46.6%)		6 (54.5%)	8 (61.5%)	
ILM-INL	5 (8.9%)	5 (8.4%)		1 (7.6%)	2 (15.3%)	0.3928	4 (9.3%)	3 (6.5%)		4 (9.3%)	4 (8.8%)		1 (9.0%)	1 (7.6%)	
IRC (65)	42 (64.6%)	32 (49.2%)	* 0.0253	13 (92.8%)	13 (92.8%)	1.0000	29 (56.8%)	19 (37.2%)	* 0.0184	30 (61.2%)	24 (48.9%)	0.1336	10 (71.4%)	7 (50.0%)	0.0833
LONLC (65)	27 (41.5%)	14 (21.5%)	* 0.0046	14 (100.0%)	6 (42.8%)	* 0.0008	13 (25.4%)	8 (15.6%)	0.1655	17 (34.6%)	8 (16.3%)	* 0.0201	9 (64.2%)	5 (35.7%)	0.1025
SRF (65)	9 (13.8%)	1 (1.5%)	* 0.0047	6 (42.8%)	1 (7.1%)	* 0.0253	3 (5.8%)	0 (0%)	0.0787	7 (14.2%)	1 (2.0%)	* 0.0143	2 (14.2%)	0 (0.0%)	0.1422
VMI (65)	41 (63.0%)	39 (60.0%)	0.3173	12 (85.7%)	11 (78.5%)	0.3173	29 (56.8%)	28 (54.9%)	0.5637	33 (67.3%)	31 (63.2%)	0.3173	8 (57.1%)	8 (57.1%)	1.0000

^1^ Eyes with positive markers (%). ^2^ McNemar’s test, comparing % of positive biomarker cases between baseline and study end for each outcome subgroup. Outer plexiform layer, OPL; outer nuclear layer, ONL; internal limiting membrane, ILM; internal nuclear layer, INL. * indicates that the p value is significant for each table (*p* < 0.05).

## Data Availability

The data generated during and analyzed in this article are available from the corresponding author, without undue reservation.

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
