# Peer review of "Optical Coherence Tomography Biomarkers in Predicting Treatment Outcomes of Diabetic Macular Edema after Ranibizumab Injections"

_medicina, 2023, doi:10.3390/medicina59030629_

Round 1
Reviewer 1 Report
Comment #1: HRF, HE, and SRF may be associated with systemic conditions including hyperlipidemia, statin-use, renal dysfunction, and hemodialysis. Please add the information about these systemic factors.
Comment #2: It is difficult to understand how to determine the OCT biomarkers, especially DRILL, LONLC, and VMI). Please show some representative OCT imaging to show how to do it.
Comment #3: Many patients who had PRP were included in this study. Acute reaction to PRP may cause macular edema by the inflammation. Please explain how to delete the acute effects of PRP.
Comment #4: Table 3. It is very surprising results that SRF may be a strong predictor for the CRT improvement after IVR injection (DR=17.586). Please discuss in detail why SRF can predict the IVR responses in patients with DME.
Comment #5: Please correct Table 5. Spacing seems incorrect.
Reviewer 2 Report
The authors present data on OCT bio markers for prediction of response of DME to ranibizumab. The manuscript has many merits, but needs significant improvement-
1. Majority of cases are PDR, with only 1 severe NPDR. Why were DME with lesser grades of DR excluded?
2. Only 22 eyes with PDR received laser photocoagulation. What was the method of management of PDR in remaining eyes? If it was anti-VEGF mono therapy- then was anti-VEGF used for PDR or for DME? As this manuscript focuses on DME, using data of PDR treated with anti-VEGF, who had co-existent DME is not correct.
3. There is no mention of number of injections received over the timeline. Mean & range is needed. Re treatment criteria need better description in methods.
4. Data states IOP change was significant. This is interesting, as IOP is of concern with steroids. Why does your cohort show decrease in IOP? Is this just incidental (most likely) - in which case is not of much relevance.
5. Data on OCT bios markers is presented well in tables, but descriptive results are minimal. It is good to highlight the significant findings in descriptive results, for ease of reader comprehension and interpretation.
6. The discussion part quotes comparative data from various publications. But it does not discuss why each biomarker is of significance. This needs to be added to the manuscript.
7. Many errors in language, grammar, repeated abbreviations, missing expansions of abbreviations- needs a do over.
